# Electrochemical cascade access to hetero[8] circulenes as potent organophotocatalysts for diverse C–X bond formations

Ahmed S. Gabr [1], Mohamed S. H. Salem [1,2] ✉, Md. Imrul Khalid[1,3], Ryota Takahashi[4], Yoshihiro Nishimoto[4], Makoto Yasuda [4] & Shinobu Takizawa [1] ✉

The chemistry of hetero[8]circulenes has been limited to three main types, constrained by synthetic challenges in creating unsymmetrical variants. Herein, we introduce an electrochemical approach to a type of hetero[8]circulene, featuring five hexagons and three pentagons. Our method capitalizes on the sustainability and selectivity of electrochemistry, utilizing differential oxidation potentials to generate dioxaza[8]circulenes through selective intermolecular and intramolecular couplings under mild conditions, achieving yields of up to 83% with good functional group tolerance. We further refine this process into a one-pot protocol using commercially available substrates, forming six new bonds. Comprehensive structural, optical, and electrochemical characterizations, including X-ray crystallography, spectrophotometric analysis, and DFT calculations, are conducted. Inspired by their distinct structural and redox properties, we explore the application of dioxaza[8]circulenes as organophotocatalysts for diverse C–X (X = C, B, S, P) bond formation achieving up to 97% yields under LED light irradiation (365 nm) without transition metals.

Hetero[*n*]circulenes are a subclass of polycyclic heteroaromatics (PHAs) distinguished by their unique structural motif of alternating carbo- and heterocyclic rings arranged in a circular fashion[1–3]. These macrocyclic molecules feature a set of [*n*] aromatic rings fused around a central polygonal core with [*n*] sides[4,5]. The incorporation of heteroatoms such as nitrogen, oxygen, or sulfur into their ring systems endows them with various desirable chemical, photophysical, and electronic properties[6–8]. In recent years, hetero[*n*]circulenes have garnered significant attention due to their promising applications in optoelectronic devices, such as organic light-emitting diodes (OLEDs)[9,10] and organic field-effect transistors (OFETs)[11] as well as various energy-related applications[12], organic photovoltaics[13], hydrogen adsorption[14,15], liquid crystals[16], and organic semiconductors[17,18].

The spectrum of hetero[*n*]circulene research spans from the smallest members, such as dithia[7]circulene, synthesized by Wynberg and co-workers[19], to larger hetero[9] and [10]circulenes synthesized independently by the Pittelkow[20], Miao[21], and Tanaka groups[22]. Among these, hetero[8]circulenes have attracted considerable interest due to the possibility to modulate π-conjugated skeletons through the incorporation of different types and numbers of heteroatoms[4,23], giving rise to intriguing electronic and optical properties[24]. Furthermore, the planar central cyclooctatetraene ring of hetero[8]circulenes presents an interesting subject of study due to its antiaromatic character[25–28].

Unlike their carbo[8]circulenes counterparts (Fig. 1a) which posed significant synthetic challenges due to their inherent strain until recent

[1]SANKEN, The University of Osaka, Ibaraki-shi, Osaka, Japan. [2]Pharmaceutical Organic Chemistry Department, Faculty of Pharmacy, Suez Canal University, Ismailia, Egypt. [3]Organic and Carbon Nanomaterials Unit, Okinawa Institute of Science and Technology Graduate University, Kunigami-gun, Okinawa, Japan. [4]Department of Applied Chemistry, Graduate School of Engineering, The University of Osaka, Suita-shi, Osaka, Japan. ✉e-mail: mohamedsalem43@sanken.osaka-u.ac.jp; taki@sanken.osaka-u.ac.jp

successes[29–31], hetero[8]circulenes have a longer history of successful preparation. The origins of hetero[8]circulene chemistry can be traced back to the late 19th century when von Knapp, Schultz, and Liebermann observed that mixing 1,4-naphthoquinones under acidic conditions produced insoluble oligomers[32,33]. This early observation laid the groundwork for future research of Erdthman and Högberg, culminating in the accurate identification of these oligomers as tetrameric structures in 1968[34,35]. Following this breakthrough, various circulenes have been synthesized using different precursors and synthetic approaches (Fig. 1). These hetero[8]circulenes can be categorized based on their structural features into three types: Type I (six hexagons and two pentagons, Fig. 1a), Type II (four hexagons and four pentagons, Fig. 1b), and Type III (eight pentagons, Fig. 1c)[4]. The construction of the central eight-membered ring is crucial, and synthetic routes are divided accordingly into two main categories: annulative construction, where the eight-membered ring is formed in the final step, and ring fusion, involving preconstructed eight-membered rings[1,23,36]. The first π-extended dihetero[8]circulene (Type I) was synthesized by Foster, Kawai, and Ito using ring fusion strategy[37], followed by other examples with different heteroatoms by Maeda and Ema (Fig. 1a)[38,39]. Unlike Type I, which has relatively few reports, Type II is the most extensively studied, utilizing both annulative and ring fusion methods[40]. Notably, Pittelkow's work involved synthesizing dioxadiaza[8]circulene and trioxaza[8]circulene via oxidative coupling of carbazole precursors[41,42]. Tanaka, and Osuka introduced tetraaza[8]circulene and triazaoxa[8]circulene via a fold-in strategy using an oxidative fusion reaction (Fig. 1b)[43,44]. The first fully heterocyclic circulene, octathia[8]circulene (sulflower) (Type III), was synthesized by Nenajdenko and co-workers from tetrathienylene (Fig. 1c)[45]. The same precursor was also utilized by Miyake to afford different hetero[8]circulens of type II using ring fusion methods[46,47].

Despite these successes, most existing methods for hetero[8]circulenes synthesis often require multi-step procedures (low total yields), harsh conditions (such as high temperature), expensive metals, and excessive oxidants (narrow functional group tolerance)[4]. These limitations restricted the current hetero[8]circulene research on using symmetrical precursors to avoid issues with chemoselectivity, and on just derivatizing already known scaffolds to fine-tune desirable characteristics, rather than attempting to break the symmetry and employ new precursors and synthetic methods to unlock a new class of compounds[46,48–50]. Inspired by the greater chemoselectivity and sustainability offered by organic electrosynthesis that emerged as an eco-friendly alternative[51–55], we leveraged our electrochemical methods, previously used for hetero[7]helicenes[56–58] and hetero[7]dehydrohelicenes[59,60] and redesigned the building blocks to synthesize an unsymmetrical type of hetero[8]circulene with five hexagons and three pentagons (Type IV). Our approach combined acid-mediated annulation of *para*-benzoquinones with 2-naphthylamine derivatives[58], followed by electrochemical coupling with 2,7-dihydroxynaphthalenes (Fig. 1d). This process, governed by the differential oxidation potentials of coupling partners, yielded dioxaza[8]circulenes through selective intermolecular and intramolecular couplings.

Herein, we have established an efficient synthesis of dioxaza[8] circulene via an electrochemical cascade process. This method was further optimized into a one-pot protocol using commercially available substrates, forming six new bonds. We conducted comprehensive structural, optical, and electrochemical characterizations for this type of hetero[8]circulene, including X-ray crystallography, spectrophotometric analysis, and DFT calculations. Inspired by the intriguing properties of the synthesized dioxaza[8]circulenes, which feature a carbazole donor with a low reduction potential and a benzo-bisbenzofuran acceptor having carrier-transport and electron

injection properties, we explored the application of hetero[n]circulenes as organophotocatalysts for various C–X bond transformations (Fig. 1d)[61–63].

## Results

### Screening of reaction conditions

To investigate the electrochemical synthesis of hetero[8]circulenes **3**, we selected hydroxybenzo[c]carbazole **1a** featuring a leaving group (X = OTf) at the 7-position, and 2,7-dihydroxynaphthalene **2** as model substrates. Based on our previous reports[56,59,64], DFT-calculations, and cyclic voltammetry (CV) studies (Supplementary Notes 1 and 8), single electron transfer (SET) from **1a** is expected to occur first, generating an electrophilic radical cation [**1a**]$^{•+}$ at the anode, as **1a** ($E_{ox}$ = 0.65 V vs. Fc/Fc$^+$ in MeCN) is more easily oxidized than **2** ($E_{ox}$ = 0.92 V vs. Fc/Fc$^+$ in MeCN). This radical cation [**1a**]$^{•+}$ rapidly undergoes deprotonation forming a neutral radical intermediate **Int-SI**, which exhibits a high spin density at the reactive position, thereby enabling subsequent intermolecular coupling with **2**. This is followed by intramolecular dehydrative cyclization to generate the corresponding helicene intermediate. Further anodic oxidation of the helicene, followed by deprotonation, furnishes another neutral radical **Int-SVI**, which undergoes C–C intramolecular coupling and final cyclization to form the dioxaza[8]circulene **3** (Supplementary Note 1).

After examining various conditions and conducting thorough optimization (see Tables S1–S3 in Supplementary Method 3), a sequential protocol was developed enabling the efficient synthesis of dioxaza[8]circulene **3a** with an 83% yield (Faradaic efficiency = 78%, Supplementary Note 2). This was achieved using platinum electrodes and tetrabutylammonium perchlorate, Bu$_4$NClO$_4$, as the supporting electrolyte in CH$_2$Cl$_2$ at 25 °C (Table 1, entry 1). Under these optimized conditions, no undesired homocoupling products were detected, highlighting the selectivity of the electrochemical approach (see Supplementary Note 3). Further investigations into the influence of reaction parameters provided deeper insights into the system's efficiency and selectivity. Increasing the current density to 1.03 mA/cm$^2$ reduced the yield of **3a** to 58% (Entry 2), suggesting that higher current densities can promote competing side reactions or lead to electrode passivation. Solvent screening revealed that CH$_2$Cl$_2$ offered the optimal balance of solubility and electrochemical stability, as other solvents led to either lower product yields or undesired byproducts (Entry 3). Additionally, CH$_2$Cl$_2$ is reported to act as a sacrificial oxidant, which can further enhance anodic oxidation and contribute to improved reaction efficiency. Electrode material was found to play a critical role in reaction efficiency. Platinum electrodes yielded the highest conversion and selectivity, outperforming combinations such as carbon/platinum, FTO (fluorine-doped tin oxide)/platinum, or FTO/FTO electrodes (Entries 4–6), due to variations in electron transfer efficiency and surface reactivity. Using alternative electrolytes, such as LiClO$_4$ or Bu$_4$NPF$_6$, decreased the yield of **3a** or did not afford the desired product (Entries 7 and 8), likely due to disrupted ion mobility and cell potential. This underscores the electrolyte's crucial role in sustaining optimal electrochemical environment. As expected, no reaction occurred in the absence of an applied potential (Entry 9), confirming the essential role of electricity in driving the transformation. Finally, to standardize the protocol and ensure reproducibility, **3a** was also synthesized in a comparable yield using ElectraSyn® 2.0 (designed by IKA) (Entry 10)[65]. To improve the atom economy of the method, it was found that using **1a'** (X = H), which does not have a leaving group, also provided the corresponding circulene in acceptable yields (Entries 11 and 12).

### Substrate scope and one-pot synthesis of hetero[8]circulenes 3

The scope of various hydroxybenzo[c]carbazole derivatives **1** was subsequently explored under the optimal reaction conditions (Fig. 2a). The facile conversion of N-aryl- and N-alkyl-substituted derivatives

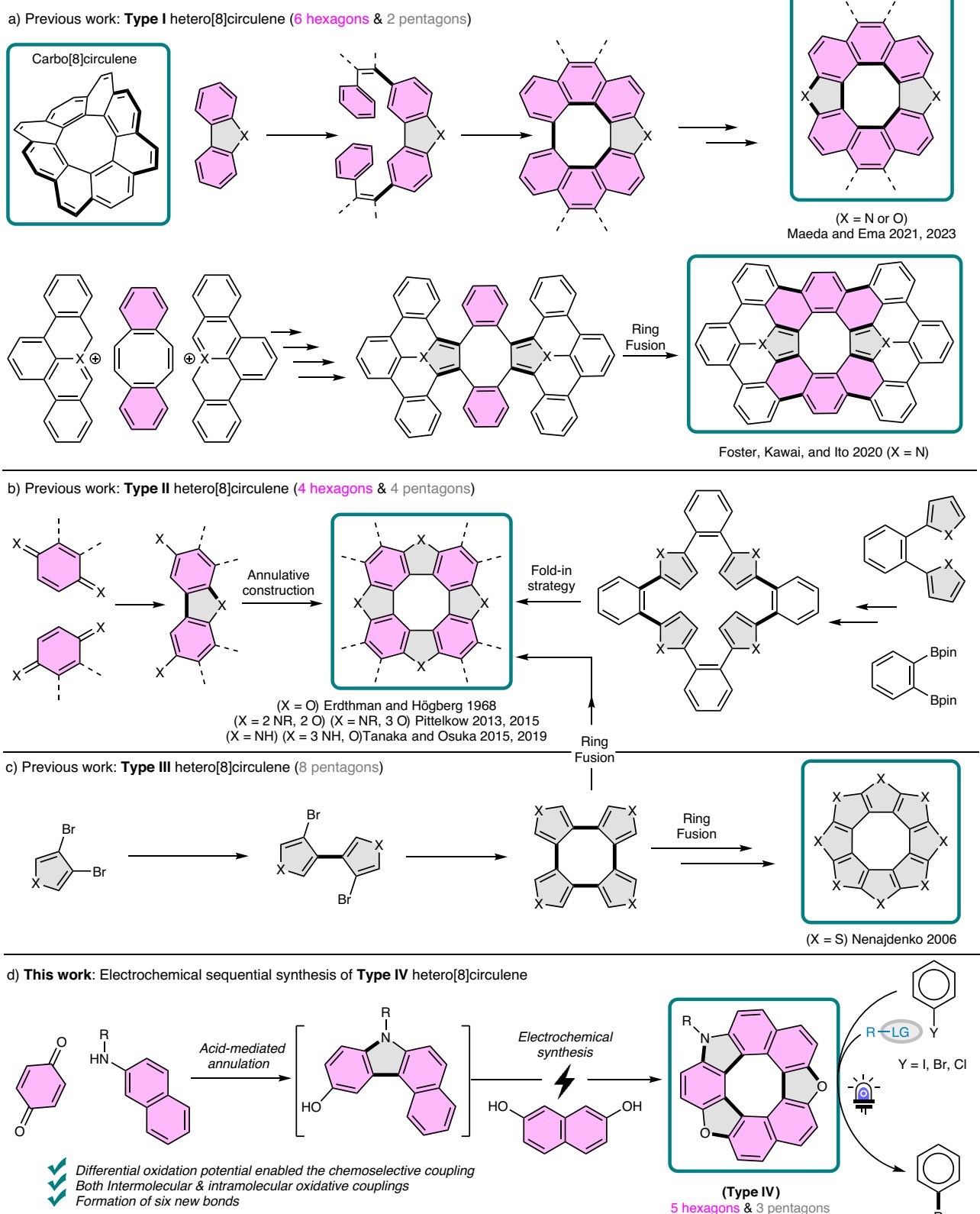

**Fig. 1 | Synthesis of different types of hetero[8]circulenes. a** Synthetic approaches to dihetero[8]circulenes (Type I, six hexagons and two pentagons); (**b**) Tetrahetero[8]circulenes (Type II, four hexagons and four pentagons); (**c**) Octahetero[8]circulenes (Type III, eight pentagons); (**d**) Electrochemical approach to trihetero[8]circulenes (Type IV, five hexagons and three pentagons) and their application as organophotocatalysts for diverse C–X bond formations; Pink ring fill highlights the hexagons, while gray ring fill highlights the pentagons; Newly formed bonds are indicated by bold lines.

**Table 1 | Optimization of reaction conditions for the electrochemical synthesis of dioxaza[8]circulene**

| Entry | Variation from standard conditions | % Yield (%FE) |
|---|---|---|
| 1 | Standard conditions using **1a** (X = OTf) | 83 (78) |
| 2 | $J = 1.03\ mA/cm^2$ | 58 (46) |
| 3 | MeCN or MeOH instead of CH$_2$Cl$_2$ | Trace |
| 4 | C(+)/Pt(−) electrodes | 38 (30) |
| 5 | FTO(+)/Pt(−) electrodes | 34 (27) |
| 6 | FTO( + )/FTO(−) electrodes | N.D. |
| 7 | Bu$_4$NPF$_6$ instead of Bu$_4$NClO$_4$ | 66 (52) |
| 8 | LiClO$_4$ instead of Bu$_4$NClO$_4$ | N.D. |
| 9 | No electricity | N.R. |
| 10 | Using ElectraSyn® | 65 (51) |
| 11 | Standard conditions using **1a'** (X = H) | 55 (39) |
| 12 | Using **1a'** (X = H) with (2.0 equiv) of **2** | 58 (41) |

Electrosynthesis conditions: Pt (Platinum) anode, Pt cathode, constant current = 1 mA, Current density ($J = 0.51\ mA/cm^2$), **1a** and **2** (0.022 mmol), Bu$_4$NClO$_4$ (2.0 mmol), CH$_2$Cl$_2$ (10 mL), 25 °C, 2.5 h; FTO: Fluorine-doped tin oxide; *N.D* not detected; *N.R* no reaction; Newly formed bonds are indicated by bold lines.

**1a–1 g** yielded dioxaza[8]circulenes **3a–3 g** in 41–83% yields. A series of **1** compounds substituted with Me, OMe, Ph, and heterocycle (thienyl) groups at different positions on the aromatic ring afforded **3h–3 m** in up to 70% yields. Both carbazoles featuring a leaving group (X = OTf) at the 7-position (**1**) and those lacking such a leaving group (**1'**) successfully underwent the cascade reaction with **2**, yielding the desired hetero[8]circulenes **3** in good-to-high yields. Carbazoles featuring functional groups such as cyano (**1n'**) and triflate (**1o'**), proved to be compatible with our conditions, affording the corresponding circulenes **3n** and **3o** in low-to-moderate yields (up to 33%). Furthermore, the π-expanded substrate **1p'** underwent the reaction with **2**, resulting in the formation of **3p** in 61% yield.

To demonstrate the practicality of this method for concise synthesis, a one-pot protocol was optimized using commercially available substrates *para*-benzoquinone **4** and *N*-aryl-2-naphthylamines **5** to form six new bonds, resulting in the synthesis of dioxaza[8]circulenes **3** in a single process (Supplementary Method 5). The acid-mediated annulation of these substrates yielded the corresponding hydroxybenzo[c]carbazole **1'** through a tandem process involving Michael addition followed by ring closure[66]. Subsequently, an electrochemical cascade reaction (comprising oxidative heterocoupling, dehydrative cyclization, intramolecular C–C bond formation, and subsequent cyclization) afforded compound **3**. This protocol proved effective for synthesizing various dioxaza[8]circulenes in overall yields of up to 35% (Fig. 2b).

### Structural, optical, and electrochemical properties of dioxaza[8]circulenes

The structure of **3a** was definitively characterized by single-crystal X-ray crystallography (Supplementary Note 7), with crystals obtained via liquid/liquid diffusion between ethyl acetate and *n*-hexane over three days in a dark environment at −20 °C (Fig. 3a). The compound's crystal structure exhibits perfect planarity, as indicated by minimal mean-plane deviation (MPD) value of 0.029 Å (Fig. 3b). The central eight-membered ring has a diameter of 3.66–3.87 Å, while the outer diameter ranges from 8.1 to 8.6 Å. The solid-state structure of **3a** reveals alternating bond lengths, with shorter radial bonds [1.39(4)–1.45(5) Å] indicating localized olefinic character, particularly in the furan [1.39(4) Å], and longer endocyclic bonds [1.42(3)–1.48(3) Å] suggesting a single bond character, consistent with a [8]radialene structure and other reported hetero[8]circulenes (Supplementary Note 10)[67]. The packing arrangement of dioxaza[8]helicenes **3** is largely dictated by multiple C − H···π interactions between neighboring molecules and displays two distinct patterns for both **3a** (cofacial lamellar packing, Figs. 3c) and **3b** (herringbone packing)[68]; further analysis of these structural features is detailed in Supplementary Notes 5 and 10.

Analyzing the isosurfaces from the anisotropy of the induced current density (AICD) plots of **3a** provides further insight into their electronic delocalization[69]. When a magnetic field is applied perpendicular to the molecular plane of **3a** (along the +z-axis, upward), conjugation through the radial C–C bonds is diminished, revealing two concentric ring currents: diatropic currents (clockwise) in the outer rim and paratropic currents (anti-clockwise) in the eight-membered core (Fig. 3d). These concentric ring currents are similar to those observed in other hetero[8]circulenes[8,70–72] and planar hetero[9]circulene[20]. Calculating the nucleus-independent chemical shift (NICS(1)$_{zz}$) values of **3a** aligns with the AICD plots, showing that all individual aromatic rings in the circulenes have negative values due to the diatropic ring current, while the central ring has positive NICS(1)zz values due to the paratropic ring current (see Supplementary Notes 6 and 10 for further discussion of aromaticity and comparison with reported hetero[n]circulenes)[73–75].

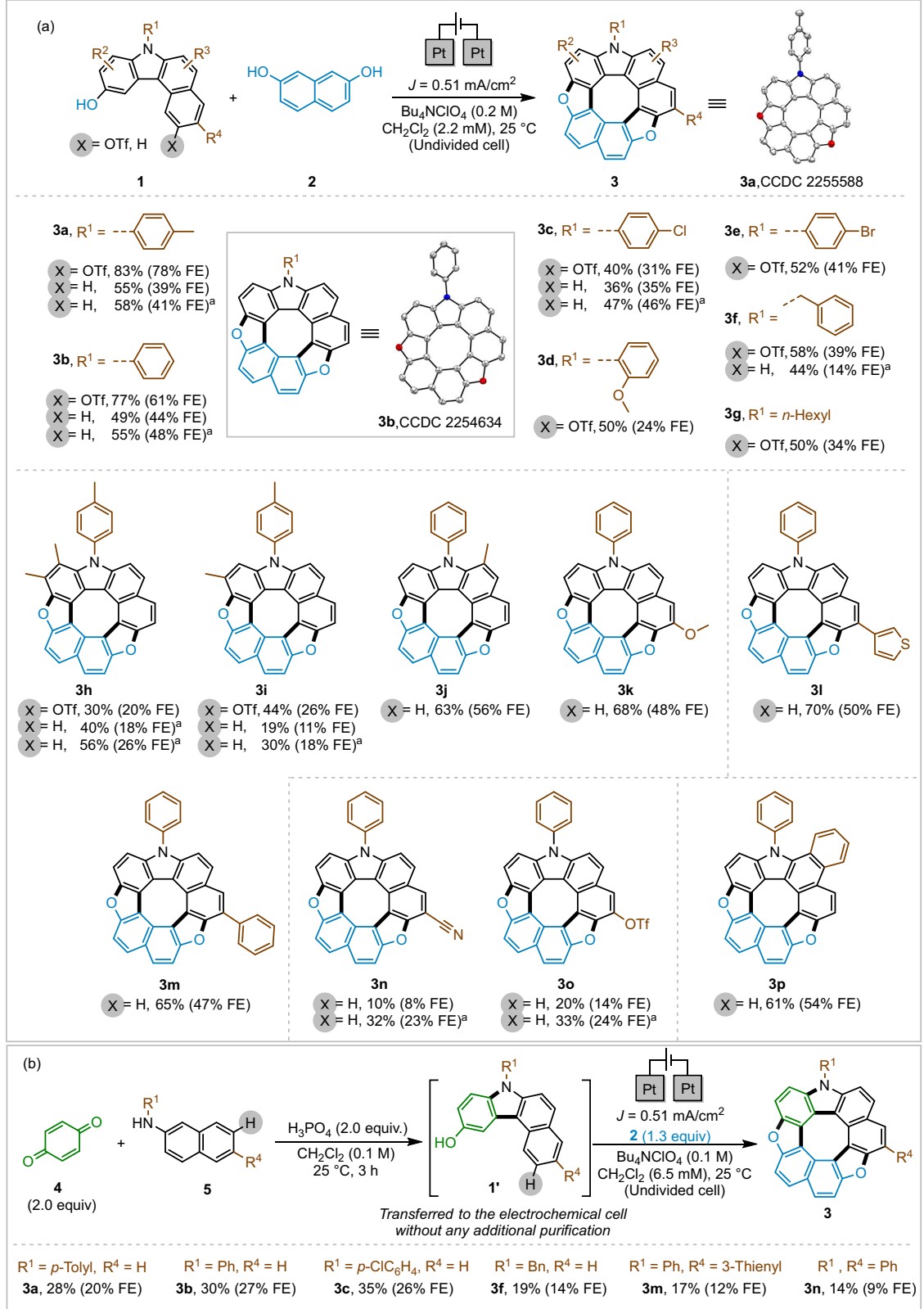

**Fig. 2 | Substrate-scope studies for the electrochemical sequential synthesis of hetero[8]circulenes. a** Electrosynthesis conditions using carbazoles [**1** (X = OTf) or **1'** (X = H)] and 2,7-dihydroxynaphthalene **2** as substrates: Pt anode, Pt cathode, constant current = 1.0 mA ($J$ = 0.51 mA/cm²), Bu₄NClO₄ (0.2 M), CH₂Cl₂ (10 mL) at 25 °C; [a]Using (2.0 equiv.) of **2**; **b** One-pot synthesis of dioxaza[8]circulenes **3** from commercially available substrates **4** and **5**; FE: Faradic efficiency; Newly formed bonds are indicated by bold lines.

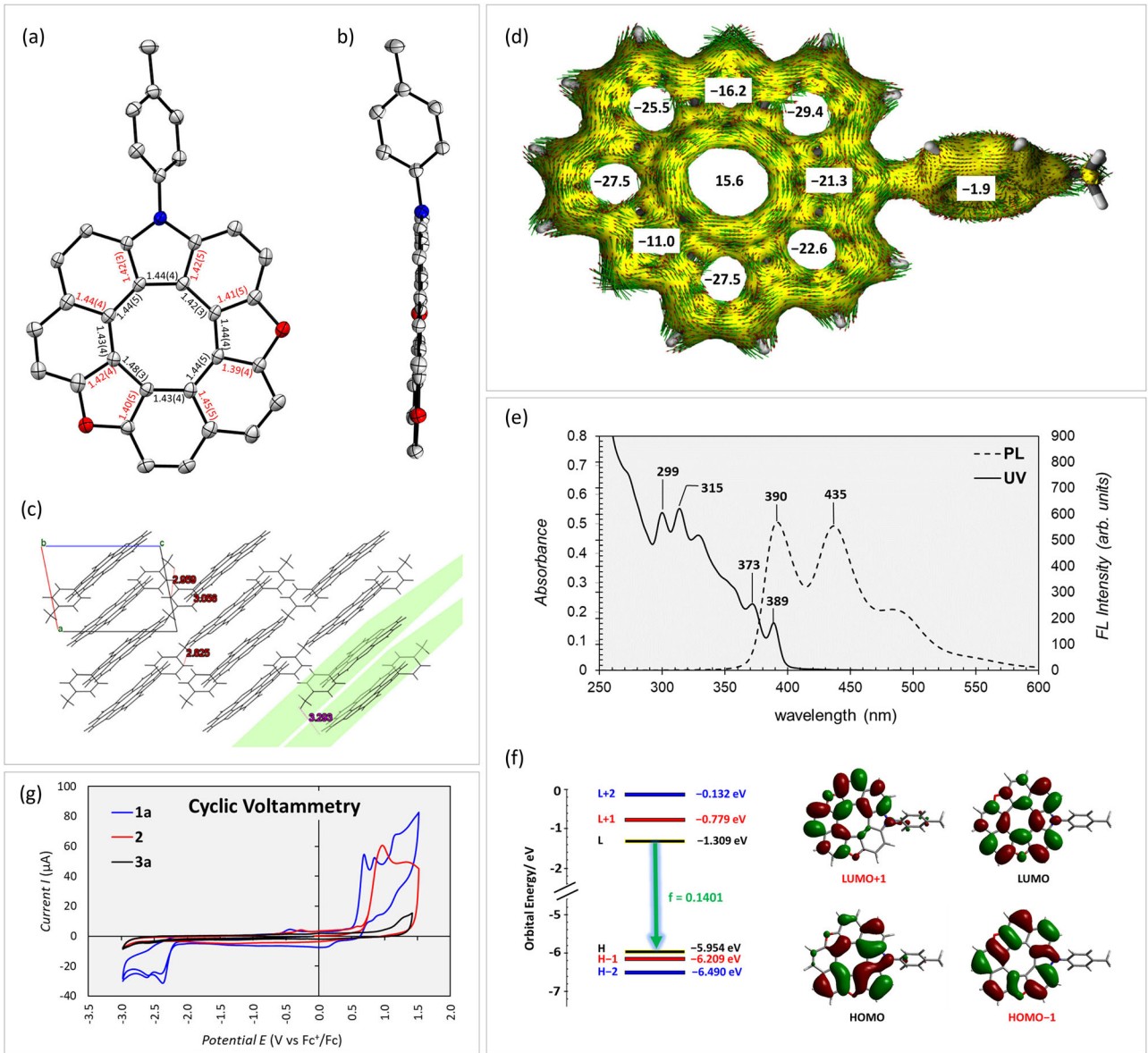

**Fig. 3 | Structural, optical, and electrochemical features of dioxaza[8] circulenes 3a. a** Top-view of the crystal structure of **3a** along with selected bond lengths illustrating the irregular shape of the aromatic rings; (**b**) Side-view of the crystal structure of **3a** with ellipsoids at 30% probability (H atoms were omitted for clarity); (**c**) Packing structure of **3a** is viewed along the b-axis to show the cofacial lamellar pattern; (**d**) Aromaticity of dioxaza[8]circulene **3a**: NICS(1)$_{zz}$ calculated at the MN15/cc-PVTZ level of theory and AICD plots calculated at the B3LYP/6-

311 G(d,p) level of theory (isosurface value: 0.05); (**e**) UV/Vis absorption and PL spectra of **3a** in chloroform (20 μM); (**f**) Frontier Kohn-Sham molecular orbitals (HOMO & LUMO) of **3a** optimized in the lowest energy excited state (S$_1$) and TD-DFT calculated transitions at MN15/ cc-PVTZ level of theory (for further information); (**g**) The cyclic voltammetry profile of **1a**, **2**, and **3a** in MeCN with $n$-Bu$_4$NClO$_4$ (0.1 M) using ferrocene as external reference.

The absorption and emission spectra of **3a** in chloroform solution (2.0 × 10$^{-5}$ M) were meticulously analyzed (Fig. 3e). Compound **3a** displayed absorption with a lower energy peak at 389 nm ($\varepsilon$ = 8.2 × 10$^3$ M$^{-1}$·cm$^{-1}$), with an optical bandgap ($E_g$) of 3.09 eV (Supplementary Note 9). The photoluminescence (PL) spectrum of dioxaza[8]circulene **3a** exhibited emission maxima at 390 and 435 nm. Time-dependent density functional theory (TD-DFT) calculations were employed to investigate the electronic transition properties, with molecular structures optimized at the lowest energy singlet excited state (S$_1$) (Supplementary Note 6)[75,76]. Frequency analysis confirmed the convergence of these optimized structures, indicating no imaginary frequencies. The frontier orbitals of dioxaza[8]circulene **3a** are non-degenerate, with only the LUMO → HOMO transitions contributing to the S$_1$ → S$_0$ transitions (Fig. 3f). The fluorescence quantum yield of **3a** in

chloroform solution (1 × 10$^{-3}$ M) is 6.5%, which can be attributed to the nonradiative transitions, particularly the intersystem crossing (ISC) rate. The redox properties of **3a** were investigated using cyclic voltammetry (CV) and differential pulse voltammetry (DPV), revealing an oxidation potential $E^{1/2}$(**3a**$^{·+}$/**3a**) of 1.34 and 1.65 V vs. SCE and a reduction potential $E^{1/2}$(**3a**/**3a**$^{·-}$) of −2.05, −2.14, and −2.46 V vs. SCE (see Fig. 3g and Supplementary Note 8).

## Application of dioxaza[8]circulene as an organophotocatalysts for diverse C−X bond formations

In recent years, the field of photoredox catalysis has witnessed a remarkable rise in prominence, primarily due to its capacity to enable transformations that were previously unattainable or considerably more challenging with traditional methodologies; the allure of

photoredox catalysis lies in its ability to generate reactive radical species under mild conditions[77–79]. While Ru(II)- and Ir(III)-based complexes are particularly esteemed for their exceptional chemical stability, extended lifetimes in the excited state, and tunability through ligand modifications[61,80], organophotocatalysts offer additional advantages such as reduced toxicity, cost-effectiveness, simplified purification, unique chemical properties, and enhanced sustainability[81]. In this domain, various examples of organocatalysts (phenothiazines[82–84], phenazines[85], phenoxazines[86], benzotriazoles[87], carbazoles[88–90], and xanthenoxanthene[91]) have been reported to function as strong photoreductants due to their high excited reductive potential (PC⁺/PC*) exceeding −1.73 V vs. SCE[92]. Inspired by the intriguing structural features of our dioxaza[8]circulene **3a**, such as the carbazole donor moiety with a low reduction potential and the benzobisbenzofuran acceptor moiety with carrier-transport and electron injection properties[62,63], as well as its redox properties with a reduction potential $E^{1/2}$(**3a**/**3a·⁻**) of −2.05 V vs. SCE and an excited reduction potential $E^{1/2}$(**3a·⁺**/**3a***) of −1.95 V vs. SCE (details in Supplementary Note 11), we envisioned exploring the applications of these circulene scaffolds as photocatalysts. Since, the reduction of organic halides is traditionally employed as a benchmark reaction to assess the reductive power of a reductant—whether for dehalogenation or for addition to electron-rich radical traps such as benzene and pyrrole—we sought to leverage this reaction to synthesize various atropisomers[91,93–95]. These axially chiral compounds have demonstrated their ubiquity across numerous domains, including functional materials, natural products, pharmaceuticals, and catalysis[96–98].

After thorough optimization of reaction conditions (see Tables S6 and S7 in Supplementary Method 6), we selected bromoacetophenone **6a** and thiophene **7a** as model substrates. Conducted in DMSO with Cs₂CO₃ and 10 mol% of dioxaza[8]circulene **3a** at room temperature under 365 nm LED light irradiation, the reaction yielded the desired product **8a** in 62%, significantly outperforming previous protocols that achieved yields no higher than 12%[95]. Various electron-rich radical traps, including furan, N-methylpyrrole, 1,3,5-trimethoxybenzene, mesitylene, and β-naphthol methyl ether, reacted with different aryl halides to produce the corresponding atropisomers **8b**–**8p** in good-to-high yields (Fig. 4). The reaction demonstrated broad functional group tolerance, accommodating both electron-withdrawing groups (e.g., cyano **8f, 8g**, and **8s**–**8y**, ester **8h**, and halide **8x**) and electron-rich aromatics (e.g., phenanthryl **8m**, naphthyl **8n**, and tolyl **8o**), yielding biaryls with radical traps in 20–91% yields. Heteroaryl halides also reacted smoothly, yielding the corresponding biaryls **8ab** at 32% yield. Aryl bromides, iodides, and occasionally chlorides were effective in this strategy, with reactivity following the trend Ar–I > Ar–Br > Ar–Cl. Expanding the scope, sulfonylation using sodium benzene sulfinates introduced sulfonyl motifs into aryls **8q**–**8u** in 37–75% yields. Borylation of aryl iodide using B₂pin₂ (3.0 equiv.) afforded **8v** in 33% yield (13% with Ar–Br). C–P bond formation was also achieved with triethyl phosphite, yielding **8y** and **8z** in 85% and 79%, respectively. Even in the absence of a base, the reaction still produced the corresponding products, albeit in lower yields.

To clarify the mechanism, control experiments were performed (Fig. 5a). The addition of tetramethylpiperidine-1-oxyl (TEMPO) inhibited the formation of **8a** (yielding only trace amounts), and the TEMPO-adduct **9** was isolated in 50% yield, indicating a radical coupling process[99]. No product was obtained without LED irradiation. To further understand the photocatalytic properties of circulene **3a**, synthesized from hydroxycarbazole **1a** and 2,7-dihydroxynaphthalene **2**, a comparative study was conducted. It was found that **1a** is the key component responsible for the photocatalytic activity of p-tolylcirculene **3a**, acting as an electron donor with strong reduction potential, whereas **2** did not yield the target compound **8** (see Supplementary Note 4). Based on the calculated redox properties of circulene **3a*** ($E^{1/2}$(**3a·⁺**/**3a***) = −1.95 V), we propose a radical coupling mechanism,

where photoreduction of the aryl halide **6** forms aryl radical **I**, which then reacts with suitable electron-rich radical traps **7** to form a new C–X bond. The radical adducts **II**, characterized by high acidity, readily undergo deprotonation in the presence of a base, generating the corresponding radical anion **III**, as substantiated by our computational pKa analysis. Subsequent oxidation of this radical anion by **3a⁺** regenerates the catalyst **3a**, thereby completing the catalytic cycle (Fig. 5b).

While recent studies have demonstrated that such radical anions **III** can act as potent reducing agents, facilitating electron catalysis[100,101], our combined experimental and computational investigations—including redox potential assessments, light on/off control experiments, and quantum efficiency measurements (~10⁻²)—strongly indicate that the predominant mechanistic pathway is primarily governed by the catalyst itself. However, the alternative electron transfer (ET) pathway, wherein radical anion **III** intermediates directly sustain the reaction by reducing **6**, cannot be definitively excluded. This secondary pathway becomes particularly relevant in some substrates where **3a** alone is insufficient to maintain catalytic turnover, yet the radical anions of specific substrates (e.g., **8m, 8n**, and **8ab**) exhibit superior reducing power, thereby facilitating the reaction (see Supplementary Note 4).

In summary, we introduced the electrochemical synthesis of a type of hetero[8]circulenes, featuring a distinctive structure of five hexagons and three pentagons. This method selectively formed unsymmetrical dioxaza[8]circulenes under mild conditions with yields up to 83%. The process was further refined into a one-pot protocol using commercially available substrates, creating six new bonds in a single process. Comprehensive structural, optical, and electrochemical analyses confirmed perfect planarity and complex electronic delocalization, with diatropic currents in the outer rim and paratropic currents in the central ring. Absorption and emission spectra, alongside TD-DFT calculations, elucidated electronic transitions, while CV and DPV revealed intriguing redox properties, including a reduction potential $E^{1/2}$(**3a**/**3a·⁻**) of −2.05 V vs. SCE and an excited reduction potential $E^{1/2}$(**3a·⁺**/**3a***) of −1.95 V vs. SCE. These features led to the application of dioxaza[8]circulenes as organophotocatalysts, achieving up to 97% yields in diverse C–X (X = C, B, S, P) bond formations.

## Methods
### General methods
For synthetic details and analytical data for all reaction products see Supplementary methods 1–7.

### General procedure for the electrochemical cascade synthesis of dioxaza[8]circulenes 3
A solution of 10-hydroxy-7H-benzo[c]carbazol-2-yl trifluoromethanesulfonate (**1**, 0.022 mmol), 2,7-dihydroxynaphthalene (**2**, 0.022 mmol), and tetrabutylammonium perchlorate (V) (2.0 mmol) in CH₂Cl₂ (10.0 mL) was transferred into an undivided electrolysis cell. This cell was equipped with two Pt electrodes (1.3 × 1.5 cm²) connected to a DC power supply. At room temperature, a constant current electrolysis with a current density of 0.51 mA/cm² was applied. After stirring for 2.5–5.0 h, the electrolysis was stopped, and the crude products were purified by column chromatography (n-hexane/EtOAc = 10/1), yielding the desired dioxaza[8]circulene **3**.

### General procedure for the one-pot synthesis of dioxaza[8]circulene 3 from commercially available starting materials
A mixture of **5** (22.0 mg, 0.1 mmol) and **4** (22.0 mg, 0.2 mmol) was dissolved in dry CH₂Cl₂ (0.5 mL). ortho-Phosphoric acid (10 μL, 0.2 mmol) in CH₂Cl₂ (0.5 mL) was then added dropwise to this mixture. The reaction was stirred for 3.0 h at 25 °C. The resulting crude mixture

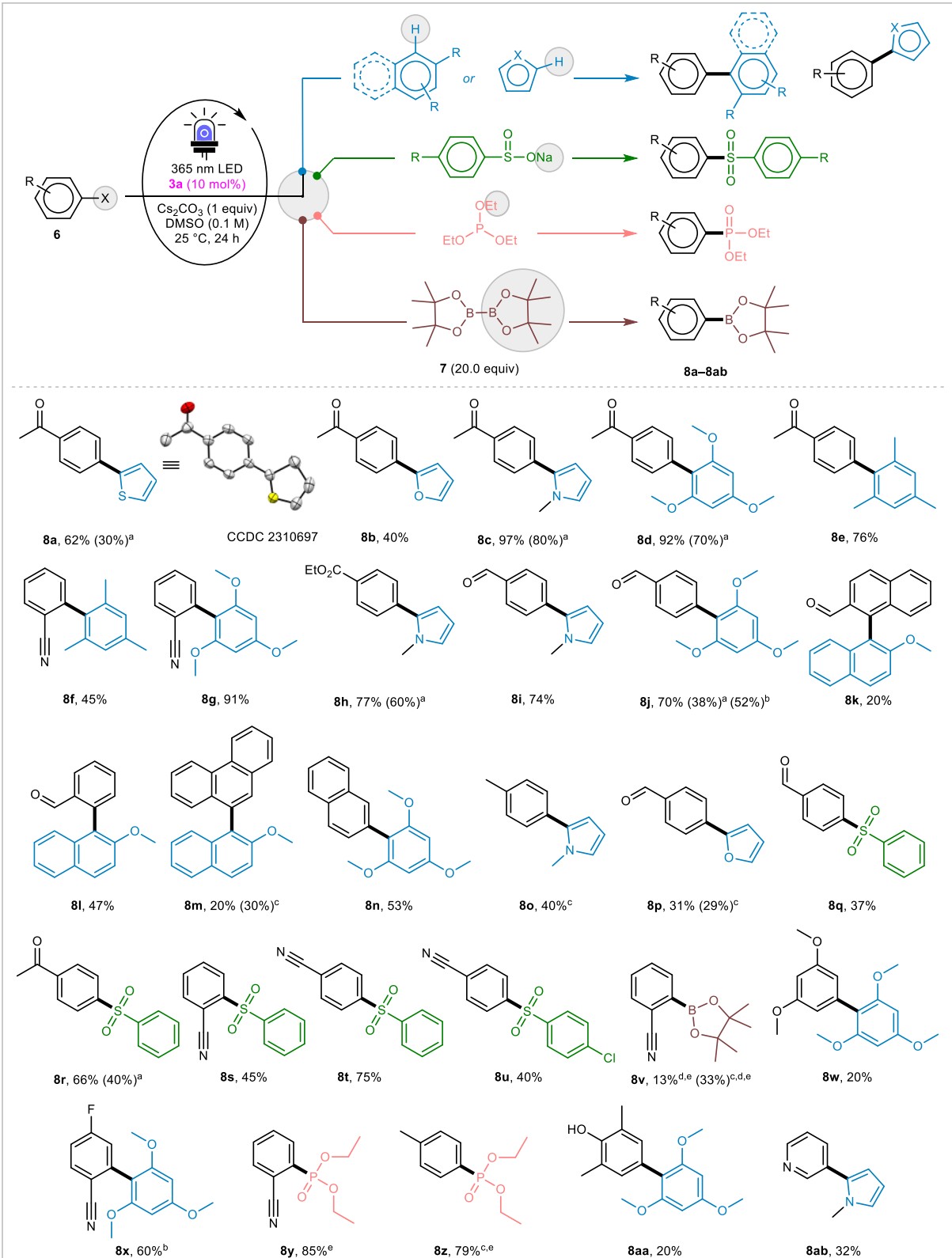

**Fig. 4 | Substrate-scope of the photocatalytic arylation reactions towards diverse C–X bond formations.** [a] Without Cs₂CO₃ (base); [b] Using Ar–Cl instead of Ar–Br; [c] Using Ar-I instead of Ar–Br; [d] Using 3.0 equiv. of **7** instead of 20.0 equiv.; [e] NMR yield; Newly formed bonds are indicated by bold lines.

was directly used in the next step without any workup. To this crude mixture, 2,7-dihydroxynaphthalene (**2**, 21 mg, 0.13 mmol, 1.3 equiv.) and tetrabutylammonium perchlorate (1.0 mmol) in CH₂Cl₂ (10 mL, 0.1 M) were added. The reaction mixture was then transferred to an undivided electrolysis cell equipped with two Pt electrodes connected to a DC power supply. At room temperature, constant current electrolysis with a current density of 0.51 mA/cm² was performed. After stirring for 12 h, the electrolysis was stopped, and the crude products were purified by column chromatography (SiO₂, n-hexane/EtOAc), yielding **3** as a yellowish white solid.

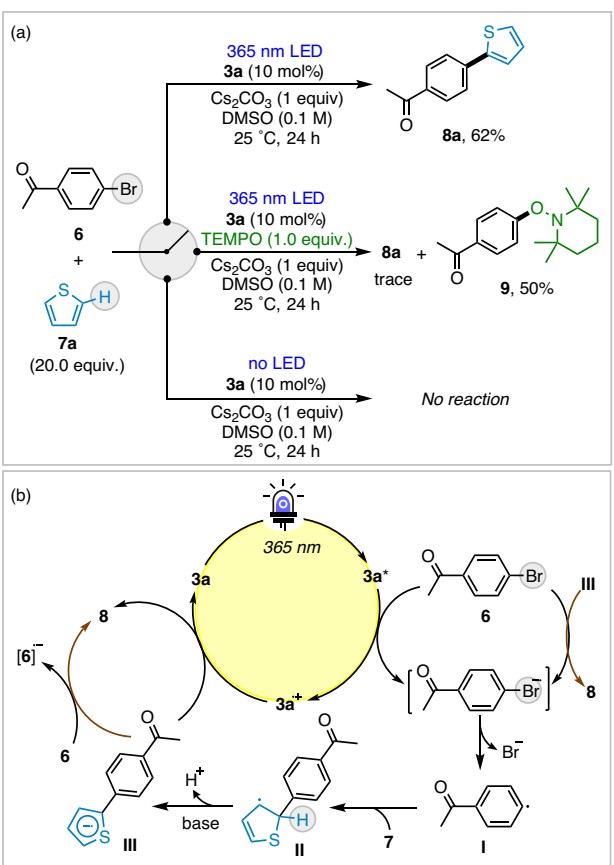

**Fig. 5 | Control experiments and plausible mechanism for the circulene-photo-catalyzed arylation reactions of 6 and 7. a** Radical inhibition and no light experiments; **b** Plausible photo-induced radical-mediated mechanistic pathway.

## General procedure for the photocatalytic arylation reactions towards diverse C-X bond formations

In a 10 mL oven-dried tube, aryl halide (0.1 mmol) **6**, dioxaza[8]circulene **3a** (0.01 mmol, 10 mol%), and $Cs_2CO_3$ (0.1 mmol, 1.0 equiv.) were added. The reaction mixture was kept under a nitrogen atmosphere, and 1.0 mL of DMSO was added, followed by the addition of (20 equiv.) of the radical trap **7**. The tube was then transferred to a water bath to maintain the temperature around 25 °C. The reaction mixture was irradiated using a 365 nm LED and stirred for 24 h. After completion, the reaction mixture was quenched with water and extracted with ethyl acetate (× 3 times). The combined organic layers were dried over anhydrous $Na_2SO_4$ and concentrated under reduced pressure. The residue was purified by column chromatography on silica gel (*n*-hexane/EtOAc) to yield the desired product **8**.

## Data availability

All data supporting the findings of this study are available within the article and its Supplementary Information. X-ray crystallographic coordinate for structures reported in this study have been deposited at the Cambridge Crystallographic Data Centre (CCDC) under the following accession codes: CCDC-2255588 [https://doi.org/10.5517/ccdc.csd.cc2fq3x6] (structure of 3a), CCDC-2254634 [https://doi.org/10.5517/ccdc.csd.cc2fp44f] (structure of 3b), and CCDC-2310697 [https://doi.org/10.5517/ccdc.csd.cc2hkgm5] (structure of 8a). These data can be obtained free of charge from The Cambridge Crystallographic Data Centre via https://www.ccdc.cam.ac.uk/structures/. The spectroscopic, electrochemical, and computational data generated in this study are also included in the Supplementary Information. Cartesian coordinates for all computed structures have been compiled into a single Excel file titled "Source Data", which is provided with this paper. No data are subject to restricted access or protected by privacy laws. All data are available from the corresponding authors upon request. Source data are provided with this paper.

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

## Acknowledgements

We acknowledge the technical staff of the Comprehensive Analysis Center of SANKEN, The University of Osaka. The computational part was performed at the Research Center for Computational Science, Okazaki (IMS-RCCS-mo-ja and IMS-RCCS-A-en) (M.S.H.S.). This work was

supported by JSPS KAKENHI grant numbers 21A204, 21H05217, 22KK0073, 22K06502 (S.T.) and 24K17681 (M.S.H.S.) from the Ministry of Education, Culture, Sports, Science, and Technology (MEXT); the Japan Society for the Promotion of Science (JSPS) and JST CREST (no. JPMJCR20R1) (S.T.).

## Author contributions

A.S.G. Designing and performing the experiments, analysing data, and writing the first draft of the manuscript. M.S. H.S. Project administration, Designing the experiments, DFT-calculations, data curation, software, visualization, writing and final editing the manuscript. M.I.K. Methodology and Validation. R.T., Y.N., and M.Y. Cyclic voltammetry (CV) and Differential pulse voltammetry (DPV) studies. S.T. Supervision, project administration, funding acquisition, and editing the final version of the manuscript.

## Competing interests

The authors declare no competing interests.
