## [Transparent Peer Review file · Nature Communications]

Electrochemical Cascade Access to Hetero[8]circulenes as Potent Organophotocatalysts for Diverse C–X Bond Formations

Corresponding Author: Professor Shinobu Takizawa

Version 0:

Reviewer comments:

Reviewer #1

(Remarks to the Author)

Hetero[n]circulenes have received considerable attentions due to their interesting electronic structures and properties, which have enabled a variety of applications. This manuscript presents a study contributing to the chemistry of hetero[8]circulenes in three key aspects: 1) the introduction of a novel type of hetero[8]circulene, featuring five hexagons and three pentagons; 2) the development of an unprecedented electrochemical approach to the synthesis of hetero[8]circulene; and 3) the innovative application of dioxaza[8]circulenes as organophotocatalysts for arylation reactions involving diverse C–X bond formations. Therefore, this manuscript is recommended for publication in Nature Communications after major revisions to address the following issues appropriately.

1. The primary concern of this reviewer is the proposed mechanism for the electrochemical synthesis of dioxaza[8]circulene (Supplementary Note 1), as detailed below:

1) The authors suggest intermolecular and intramolecular coupling reactions through neutral radical intermediates (Int-SI, Int-SII, Int-SIII; Int-SIX, Int-SX, Int-SXI). However, these coupling reactions could, in principle, also occur through radical cation intermediates, similar to the Scholl reaction. The authors should provide evidence, either experimental or computational, to support that the electrochemical synthesis proceeds via a neutral radical pathway rather than a radical cation pathway.

2) Int-SI and Int-SII are resonance structures, and the transition from Int-SI to Int-SII is not a single electron transfer (SET) process. In contrast, an electron transfer process is involved in the transition from 1 to Int-SI. Similarly, the transition from Int-SIX to Int-SX is not an SET process either.

3) Importantly, the final step in the proposed mechanism cannot explain the reaction of 1a', which lacks a leaving group, nor the one-pot synthesis of 3 from compound 5, which also lacks a leaving group.

2. Page 8 states that "single electron transfer (SET) from 1a is expected to occur first," and "intermolecular radical-anion coupling between the radical species and 2." To ensure that SET occurs only on compound 1 rather than 2 (Supplementary Note 1), the anode potential should be higher than the oxidation potential of 1 but lower than that of 2. The electrochemical reaction was conducted with a constant current, $J = 0.51 \text{ mA/cm}^2$. During the process, was the anode potential maintained within the required range? Additionally, the oxidation potentials of 1 and 2 should be reported in volts (V) and referenced to a standard (e.g., Fc+/Fc).

3. The CV and DPV charts in Supplementary Note 7 are problematic. The potential in all the CV charts is reported as "vs Fc+/Fc". Therefore, the oxidation potential of ferrocene must be 0 because it is versus itself. How could the potential for ferrocene be 0.0795 V? With this error, the reported potentials for other compounds versus Fc+/Fc are all incorrect. Furthermore, the potentials in the DPV of 1a, 2a, and 3a should also be reported versus a reference (e.g., Fc+/Fc).

Reviewer #2

(Remarks to the Author)

This manuscript provides an electrochemical approach to a novel type of hetero[8]circulene and refined it into a one-pot protocol. Furthermore, unprecedented application of dioxaza[8]circulenes as organophotocatalysts was explored for diverse C–X (X = C, B, S, P) bond formation in high yield. This paper is sloppy in many places, which does not meet the high requirements of Nature Communications, some details in this paper need to be added or modified, specifically as follows:

1. The entire manuscript needs to be carefully polished. The present text seems like a report, the author should carefully discuss the results of the screening of reaction conditions, and explain the selectivity of electrochemistry.
2. Please provide clear ¹H NMR spectra of all the compound in SI, and integrate them correctly.

Reviewer #3

(Remarks to the Author)

Takizawa and coworkers report an electrochemical cascade coupling paving access to a variety of hetero[8]circulenes, structures previously accessed through Lewis-acidic couplings. To the best of my knowledge, anodic coupling has not been used to access this compound class so far. While numerous polycyclic (hetero)aromatics have been utilized as photocatalysts, it seems that heterocirculenes have been spared so far in these investigations so that both aspect of the presented work are indeed novel. The photoexcited catalyst appears to be strong enough a reductant that it cleaves the C–X bond in acceptor substituted bromoarenes. Strangely, biaryl couplings can also be run with substrates, the redox potentials of which lie out of reach for the catalyst so that alternative explanations need to be taken into consideration. The recently investigated deprotonations of the intermediate radicals to strongly reducing radical anions, which permit electron catalysis to take place, might be an option and would also explain why the presence of the base plays a role (which it should not in the proposed mechanism).

Outside of the mechanistic considerations, the paper is almost flawless (for some minor points, see below) and it definitely deserves disclosure in a good or very good journal. Whether or not the paper reaches the bar here is of course at the editor's discretion but this reviewer has no objections and supports publication after a round of revisions.

Minor points:

- Please consider the commented pdf attached.
- Another manuscript, which should be cited in the context of the aromaticity analysis is <https://doi.org/10.1002/poc.4609>, where various hetero[8]circulenes have recently been investigated.
- Supporting information: For known solid compounds (e.g. the biaryl-type coupling products), melting point ranges should be given alongside the respective literature values.
- SI, p.58: Wavelength
- SI: Some of the NMR spectra contain significant aliphatic signals (e.g. 1l', 3a, 8k, etc.), which shouldn't be there. These should be removed, e.g. by a pentane wash followed by drying in vacuo.

Version 1:

Reviewer comments:

Reviewer #1

(Remarks to the Author)

The authors have appropriately addressed all the comments of this reviewer mainly in the Supplementary Information. The revised mechanism based on DFT-calculation is convincing. This manuscript is almost ready for publication except the following issue.

It is stated on Page 8 that " After intermolecular radical-anion coupling between the radical species and 2, followed by two consecutive cyclizations and intramolecular coupling, dioxaza[8]circulene 3a can be obtained (Supplementary Note 1)." This statement does not correctly summarize the reaction mechanism studies presented in the Supplementary Note 1. The key coupling reaction is between compound 2 and radical Int-SI, which is not a radical-anion coupling. Furthermore, the current discussion on the reaction mechanism in the main text is too brief. The authors are strongly suggested to add a few sentences to the main text to summarize their results in the Supplementary Note 1.

Reviewer #2

(Remarks to the Author)

Shinobu Takizawa and co-author have revised the manuscript according to reviewer's comments, therefore, after the authors check the text and data format, I suggest the editor to accept the manuscript.

Reviewer #3

(Remarks to the Author)

Takizawa and coworkers have thoroughly revised their manuscript according to the reviewer comments and have significantly improved its quality. They have done extensive calculations on the proposed neutral radical pathway for the closure of the macrocycle as well as for the radical anion pathway, they have scrutinized the potential referencing, have added a light on/off experiment to exclude slow radical chains, have improved the purity of the products, have included melting point ranges and have corrected some minor issues.

I only have two more points to raise. Several of the calculated energy barriers are very high, sometimes prohibitively high, so that the reactions should not occur. For instance, the deprotonation of the phenol radical cation through TS-IX seems to have

an enormous barrier. Not sure why that is – most certainly, a perchlorate ion will accept the proton in a hydrogen-bonding situation, which (hopefully) dramatically drives down the barrier to a value that makes sense physically. The subsequent transition state energies are also very high and this reviewer wonders, whether these “simple” reactions take another course. The H-atom abstraction depicted in TS-XI is most likely inferior to a second oxidation/deprotonation. Is TS-XII an electron transfer? If so, how to calculate a meaningful barrier for this step? Free electron moving through space? Marcus theory? In the Scheme on page 54 of the SI, the representations of the radical Int-SI need to be connected by a mesomeric arrow, not by an equilibrium arrow. They are the same structure, just showing the spin densities on different centers. The paper is otherwise in very good shape, very detailed and will certainly attract the readers of Nature Communications. I strongly recommend acceptance after a final round of revisions.

Version 2:

Reviewer comments:

Reviewer #1

(Remarks to the Author)

The authors have appropriately addressed all the comments of this reviewer appropriately. The revised manuscript is recommended for publication in Nature Communications.

Reviewer #3

(Remarks to the Author)

Takizawa and coworkers have further revised their manuscript according to the reviewer comments and have once again improved its quality, also by including the counterion in the calculations of their DFT reaction barriers, which are now closer to a reasonable range. Moreover, they have calculated energies for a new pathway, which turned out to be lower in barrier height than the other processes previously investigated. While this still doesn't have to be the mechanistic truth, it represents a reasonable assumption.

Overall, I congratulate the authors to these interesting results and support acceptance of the manuscript without further change.

Reviewers' comments:

Reviewer 1 comments

Hetero[*n*]circulenes have received considerable attentions due to their interesting electronic structures and properties, which have enabled a variety of applications. This manuscript presents a study contributing to the chemistry of hetero[8]circulenes in three key aspects: 1) the introduction of a novel type of hetero[8]circulene, featuring five hexagons and three pentagons; 2) the development of an unprecedented electrochemical approach to the synthesis of hetero[8]circulene; and 3) the innovative application of dioxaza[8]circulenes as organophotocatalysts for arylation reactions involving diverse C–X bond formations. Therefore, this manuscript is recommended for publication in Nature Communications after major revisions to address the following issues appropriately.

Reviewer 1-1: The primary concern of this reviewer is the proposed mechanism for the electrochemical synthesis of dioxaza[8]circulene (Supplementary Note 1), as detailed below:

The authors suggest intermolecular and intramolecular coupling reactions through neutral radical intermediates (Int-SI, Int-SII, Int-SIII; Int-SIX, Int-SX, Int-SXI). However, these coupling reactions could, in principle, also occur through radical cation intermediates, similar to the Scholl reaction. The authors should provide evidence, either experimental or computational, to support that the electrochemical synthesis proceeds *via* a neutral radical pathway rather than a radical cation pathway.

Our response: We deeply appreciate the reviewer's insightful comment and have carefully investigated this aspect through computational analysis, considering energy profiles, spin density distributions, pKa values, and redox behaviors of all intermediates. Based on these studies, we have revised and included more details in the proposed mechanism to address the reviewer's concerns comprehensively (see Supplementary Note 1, where we updated the mechanism and included our computational investigation of different reaction pathways).

Earlier studies by Siegfried R. Waldvogel {*Synthesis* **49**, 252–259 (2017), *Synlett* **30**, 2062–2067 (2019), *Acc. Chem. Res.* **53**, 45–61 (2019), *Chem. Eur. J.* **25**, 2713–2716 (2019), *Angew. Chem. Int. Ed.* **53**, 5210–5213 (2014)}, and our DFT calculations indicated that the chemoselective oxidation of carbazole **1** occurs at the anode, driven by its lower oxidation potential relative to 2,7-dihydroxy naphthalene **2**. The oxidation step is **immediately**

followed by deprotonation, as the radical cation $[1]^{+\bullet}$ is highly acidic. Our computed pKa values further supported this hypothesis, reinforcing the role of radical cation intermediates in the reaction pathway.

$$pK_a = \frac{G_{sol}(H^+) + G_{sol}(\text{conjugate base}^-) - G_{sol}(\text{Acid})}{2.3 RT}$$

Molecule	$E_{(\text{total})}$ Hartree	$E_{(\text{total})}$ kcal/mol	ΔG (kcal/mol)	pKa
1b'-OH	-977.18835	-613194.4843	31.65155396	23.2
1b'-O⁻	-976.70892	-612893.6377		
[1b']^{•+}	-976.98605	-613067.5392	-7.56775854	-5.6
[1b'][•]	-976.56912	-612805.9119		

Optimized at the UB3LYP/6-31G+(d,p) level of theory with IEPCM model as solvation of DCM. Grimme's dispersion with the original D3 damping function was applied as empirical dispersion correction to the optimized structures.

✚ Once these radical cations $[1]^{+\bullet}$ are generated, two possible pathways emerge: either proton release due to their high acidity ($pK_a \sim -5.6$) or radical-anion coupling with 2,7-dihydroxynaphthalene **2**. To investigate this, we analyzed the spin densities of both the radical cation species $[1]^{+\bullet}$ and their corresponding neutral radicals **Int-SI** after proton release. These studies revealed a significant difference in spin density at the key reactive position, suggesting that the radical cation $[1]^{+\bullet}$ cannot directly undergo coupling with **2**. Instead, it must first convert into the corresponding intermediate, **Int-SI**, before proceeding with the reaction.

Mulliken spin density contour maps calculated at the UB3LYP/6-31G+(d,p)/IEFPCM=DCM. Grimme's dispersion with the original D3 damping function was applied as empirical dispersion correction to the optimized structures. (isoval = 0.003).

We have added more data for the helicenes and all intermediates included in the mechanistic cycle. In addition, we calculated the energy profiles of all intermediates and transition states in possible pathways to conclude the most energetically favorable path and support our plausible mechanism.

Reviewer 1-2: Int-SI and Int-SII are resonance structures, and the transition from Int-SI to Int-SII is not a single electron transfer (SET) process. In contrast, an electron transfer process is involved in the transition from **1** to Int-SI. Similarly, the transition from Int-SIX to Int-SX is not an SET process either.

Our response: We have carefully revised our mechanism, correcting all aspects related to resonance structures and SET processes.

Reviewer 1-3: Importantly, the final step in the proposed mechanism cannot explain the reaction of **1a'**, which lacks a leaving group, nor the one-pot synthesis of **3** from compound **5**, which also lacks a leaving group.

Our response: We have reconsidered the mechanism and propose that two additional anodic events can occur to facilitate the subsequent C–O insertion. This revised mechanism aligns well with our energy profile study, pKa, E_{ox}, and spin density analysis. According to this update, we have also revised the %FE calculations of methods B and C.

Calculated at the UB3LYP/6-31G+(d,p)/IEFPCM=DCM. Grimme's dispersion with the original D3 damping function was applied as empirical dispersion correction to the optimized structures. All global minima were verified by the absence of imaginary frequencies, while all transition states were confirmed by the presence of a single imaginary frequency. Additionally, an Intrinsic Reaction Coordinate (IRC) analysis was performed to ensure that each transition state correctly connects the corresponding reactant and product minima along the reaction pathway

Reviewer 1-4: Page 8 states that “single electron transfer (SET) from **1a** is expected to occur first,” and “intermolecular radical-anion coupling between the radical species and **2**.” To ensure that SET occurs only on compound **1** rather than **2** (Supplementary Note 1), the anode potential should be higher than the oxidation potential of **1** but lower than that of **2**. The electrochemical reaction was conducted with a constant current, J = 0.51 mA/cm². During the process, was the anode potential maintained within the required range? Additionally,

the oxidation potentials of **1** and **2** should be reported in volts (V) and referenced to a standard (e.g., Fc⁺/Fc).

Our response: We have monitored the potential of the working electrode over 1 hour, which was higher than the oxidation potential of **1** and lower than that of **2**, confirming selective oxidation (see Supplementary Note 3). Furthermore, our investigations and DFT calculations showed that all generated intermediates and helicenes have oxidation potentials lower than the working electrode potential, explaining the observed selectivity. Additionally, we have now reported the oxidation potentials of **1** and **2** in volts (V) and referenced them to Fc⁺/Fc in the manuscript.

Reviewer 1-5: The CV and DPV charts in Supplementary Note 7 are problematic. The potential in all the CV charts is reported as “vs Fc⁺/Fc”. Therefore, the oxidation potential of ferrocene must be 0 because it is versus itself. How could the potential for ferrocene be 0.0795 V? With this error, the reported potentials for other compounds versus Fc⁺/Fc are all incorrect. Furthermore, the potentials in the DPV of **1a**, **2a**, and **3a** should also be reported versus a reference (e.g., Fc⁺/Fc).

Our response: We apologize for the previous misleading report. The uncorrected oxidation potential of ferrocene was initially included (0.0795) to illustrate how we corrected the CV charts. However, all CV and DPV charts for **1**, **2**, and **3** have already been properly corrected using Fc⁺/Fc (set to 0 V). To avoid confusion, we have now clearly labeled the uncorrected value and provided both the corrected and uncorrected CV charts for better clarity and understanding (see supplementary Note 8).

Reviewer 2 comments

This manuscript provides an electrochemical approach to a novel type of hetero[8]circulene and refined it into a one-pot protocol. Furthermore, unprecedented application of dioxaza[8]circulenes as organophotocatalysts was explored for diverse C–X (X = C, B, S, P) bond formation in high yield. This paper is sloppy in many places, which does not meet the high requirements of Nature Communications, some details in this paper need to be added or modified, specifically as follows:

Reviewer 2-1: The entire manuscript needs to be carefully polished. The present text seems like a report, the author should carefully discuss the results of the screening of reaction conditions, and explain the selectivity of electrochemistry.

Our response: We appreciate reviewer 2's feedback and insights. In the revised manuscript, we have carefully polished the text and added details to further explain and discuss the results of the reaction condition screening and the selectivity of electrochemistry. These revisions provide a clearer analysis of the key factors influencing efficiency and selectivity.

Reviewer 2-2: Please provide clear ¹H NMR spectra of all the compound in SI, and integrate them correctly.

Our response: In response to the reviewer's suggestion, we have provided clearer ¹H NMR spectra with enhanced resolution, including zoomed-in or enlarged peaks for improved readability (see supplementary Data). Additionally, we have further purified several compounds, including 8g, 8k, 8h, 8m, 8u, 8v, 1a, 11', and 3a, among others, which previously exhibited aliphatic or solvent impurities. Their spectra were re-measured to ensure high purity and accurate integration.

Reviewer 3 comments

Takizawa and coworkers report an electrochemical cascade coupling paving access to a variety of hetero[8]circulenes, structures previously accessed through Lewis-acidic couplings. To the best of my knowledge, anodic coupling has not been used to access this compound class so far. While numerous polycyclic (hetero)aromatics have been utilized as photocatalysts, it seems that heterocirculenes have been spared so far in these investigations so that both aspect of the presented work is indeed novel.

Reviewer 3-1: The photoexcited catalyst appears to be strong enough a reductant that it cleaves the C-X bond in acceptor substituted bromoarenes. Strangely, biaryl couplings can also be run with substrates, the redox potentials of which lie out of reach for the catalyst so that alternative explanations need to be taken into consideration. The recently investigated deprotonations of the intermediate radicals to strongly reducing radical anions, which permit electron catalysis to take place, might be an option and would also explain why the presence of the base plays a role (which it should not in the proposed mechanism).

Our response: Thank you for the insightful comment. Inspired by this suggestion, we conducted a detailed mechanistic investigation to evaluate the proposed pathway involving radical anions. Our energy profile studies confirmed that this pathway is energetically feasible.

However, our DFT calculations indicate that most generated radical anions have lower reducing power than the catalyst. Additionally, light/dark experiments and quantum efficiency calculations do not strongly support a chain reaction *via* direct electron transfer.

That said, the mechanism cannot be entirely ruled out, as for certain substrates, the catalyst alone lacks sufficient reducing power to sustain the reaction, whereas their radical anions exhibit significantly stronger reducing capabilities (e.g., **8m**, **8n**, and **8ab**). Based on this, we have revised the mechanism to include radical anion formation and clarified that while the catalyst-driven pathway dominates for most substrates, in some cases, the alternative pathway plays a more active role. These clarifications have been incorporated into the manuscript, and the computational and experimental studies have been detailed in Supplementary Note 4.

Reviewer 3-2: Outside of the mechanistic considerations, the paper is almost flawless (for some minor points, see below) and it definitely deserves disclosure in a good or very good journal. Whether or not the paper reaches the bar here is of course at the editor's discretion but this reviewer has no objections and supports publication after a round of revisions. Please consider the commented pdf attached.

Our response: We have carefully revised the manuscript, addressing all points raised in the attached file.

Reviewer 3-3: Another manuscript, which should be cited in the context of the aromaticity analysis is <https://doi.org/10.1002/poc.4609>, where various hetero[8]circulenes have recently been investigated.

Our response: Thank you for the suggestion. We have now cited Reference 72 in the context of the aromaticity analysis.

Reviewer 3-4: Supporting information: For known solid compounds (e.g. the biaryl-type coupling products), melting point ranges should be given alongside the respective literature values.

Our response: We have included the melting point ranges for all known solid compounds, such as the biaryl-type coupling products, alongside the respective literature values in the Supporting Information as indicated by Reviewer 3.

Reviewer 3-5: SI, p.58: Wavelength

Our response: We corrected the spelling of this word.

Reviewer 3-6: SI: Some of the NMR spectra contain significant aliphatic signals (e.g. 11', 3a, 8k, etc.), which shouldn't be there. These should be removed, e.g. by a pentane wash followed by drying in vacuo.

Our response: We have carefully reviewed all compounds and further purified most of them, including 11', 8k, 8h, 8m, 8u, 8v, 1a, 1c, and 3a, 3c, among others, to remove aliphatic impurities and solvent peaks. The updated ¹H NMR spectra in the Supporting Information reflect these improvements, with enhanced resolution and zoomed-in or enlarged peaks for better readability, confirming the enhanced purity of the compounds.

Reviewers' comments:

Reviewer 1 comments

The authors have appropriately addressed all the comments of this reviewer mainly in the Supplementary Information. The revised mechanism based on DFT-calculation is convincing. This manuscript is almost ready for publication except the following issue.

Reviewer 1-1: It is stated on Page 8 that " After intermolecular radical-anion coupling between the radical species and 2, followed by two consecutive cyclizations and intramolecular coupling, dioxaza[8]circulene 3a can be obtained (Supplementary Note 1)." This statement does not correctly summarize the reaction mechanism studies presented in the Supplementary Note 1. The key coupling reaction is between compound 2 and radical **Int-SI**, which is not a radical-anion coupling. Furthermore, the current discussion on the reaction mechanism in the main text is too brief. The authors are strongly suggested to add a few sentences to the main text to summarize their results in the Supplementary Note 1.

Our response: We thank the reviewer for pointing this out. In response, we have removed the inaccurate phrase describing the reaction as a "radical-anion coupling" and have revised the mechanistic section in the main text to provide a more accurate and informative summary of our findings from Supplementary Note 1. Specifically, we now describe the key steps involving the generation of a radical cation from **1a**, its deprotonation to form the neutral radical intermediate **Int-SI**, and the subsequent coupling with compound **2**. We also detail the formation of the helicene intermediate and its progression through further oxidation, deprotonation, and intramolecular cyclization steps to furnish the final product **3a**.

The revised paragraph in the main text (Page 8) now reads:

"Based on our previous reports, DFT-calculations, and cyclic voltammetry (CV) studies (Fig. 3g, and Supplementary Notes 1 and 8), single electron transfer (SET) from 1a is expected to occur first, generating an electrophilic radical cation [1a]^{•+} at the anode, as 1a ($E_{ox} = 0.65$ V vs. Fc/Fc⁺ in MeCN) is more easily oxidized than 2 ($E_{ox} = 0.92$ V vs. Fc/Fc⁺ in MeCN). This radical cation [1a]^{•+} rapidly undergoes deprotonation forming a neutral radical intermediate **Int-SI**, which exhibits a high spin density at the reactive position, thereby enabling subsequent intermolecular coupling with 2. This is followed by intramolecular dehydrative cyclization to generate the corresponding helicene intermediate. Further anodic oxidation of the helicene, followed by deprotonation, furnishes another neutral radical **Int-SVI**, which undergoes C–C intramolecular coupling and final cyclization to form the dioxaza[8]circulene 3 (Supplementary Note 1)."

We hope this improved explanation sufficiently addresses the reviewer's concern.

Reviewer 2 comments

Shinobu Takizawa and co-author have revised the manuscript according to reviewer's comments, therefore, after the authors check the text and data format, I suggest the editor to accept the manuscript.

Our response: We sincerely thank the reviewer for their positive assessment and kind recommendation for acceptance. We have carefully re-checked the text and data format to ensure consistency and accuracy throughout the manuscript and supporting information.

Reviewer 3 comments

Takizawa and coworkers have thoroughly revised their manuscript according to the reviewer comments and have significantly improved its quality. They have done extensive calculations on the proposed neutral radical pathway for the closure of the macrocycle as well as for the radical anion pathway, they have scrutinized the potential referencing, have added a light on/off experiment to exclude slow radical chains, have improved the purity of the products, have included melting point ranges and have corrected some minor issues. I only have two more points to raise:

Reviewer 3-1: Several of the calculated energy barriers are very high, sometimes prohibitively high, so that the reactions should not occur. For instance, the deprotonation of the phenol radical cation through TS-IX seems to have an enormous barrier. Not sure why that is – most certainly, a perchlorate ion will accept the proton in a hydrogen-bonding situation, which (hopefully) dramatically drives down the barrier to a value that makes sense physically. The subsequent transition state energies are also very high and this reviewer wonders, whether these “simple” reactions take another course.

Our response: We thank the reviewer for this valuable observation and helpful suggestion. In response, we have revised the calculations of all deprotonation steps and explicitly included the perchlorate ion to model the proton transfer. This modification significantly reduced the associated energy barriers to more physically reasonable values. Additionally, we re-optimized all intermediates and transition states in the main and side pathways and, where appropriate, employed QST-2 calculations to ensure accurate reaction profiles. While the general mechanistic pathway remains unchanged, the revised energy profiles now provide a much more convincing and realistic depiction of the reaction sequence. Importantly, the revised barrier trends now align well with the relative pKa values of the intermediates, further supporting the updated mechanism. All relevant sections and Cartesian coordinates in the revised Supporting Information (Supplementary Note 1) have been updated accordingly.

Calculated at the UB3LYP/6-31G+(d,p)/IEFPCM=DCM. Grimme's dispersion with the original D3 damping function was applied as empirical dispersion correction to the optimized structures. All global minima were verified by the absence of imaginary frequencies, while all transition states were confirmed by the presence of a single imaginary frequency. Additionally, an Intrinsic Reaction Coordinate (IRC) analysis was performed to ensure that each transition state correctly connects the corresponding reactant and product minima along the reaction pathway.

Reviewer 3-2: The H-atom abstraction depicted in **TS-XI** (**TS-XII*** in the new version) is most likely inferior to a second oxidation/deprotonation. Is **TS-XII** (**TS-XIV** in the new version) an electron transfer? If so, how to calculate a meaningful barrier for this step? Free electron moving through space? Marcus theory?

Our response: We fully agree that a second oxidation followed by deprotonation could present a more favorable pathway than the previously proposed H-atom abstraction. In fact, this alternative was briefly mentioned in our earlier version, where we suggested it as a plausible scenario based on the predicted pKa and redox properties of the intermediates, although calculations were not included at that time. In the revised manuscript, we have now explicitly evaluated this pathway. Not only for this step, but also for all similar steps in our mechanism (Int-SIIa, Int-SVII, and Int-SX). We modeled the electron transfer step using Marcus theory to estimate a meaningful activation barrier and employed a perchlorate anion to assist the subsequent deprotonation step. This revised pathway was found to be less energetically demanding and overall more favorable than the H-abstraction route.

Additionally, we have added a new section in Supplementary Note 1 detailing our use of the four-points' Buda method, which provides reliable estimates of reorganization energies ROEs in the electrochemical reactions. All relevant data and Cartesian coordinates have been updated accordingly in the revised Supporting Information.

$$\Delta G^\ddagger = \frac{(\Delta G^\circ + \lambda)^2}{4\lambda}$$

	1b'	Int-SIIa ^o	Hel	Int-SVII ^o	Int-SVIII	Int-SX ^o
λ_{\rightarrow} (Hartree)	0.025	0.028	0.023	0.026	0.024	0.023
λ_{\leftarrow} (Hartree)	0.027	0.028	0.023	0.026	0.024	0.024
$\lambda_{\leftrightarrow}$ (Hartree)	0.026	0.028	0.023	0.026	0.024	0.024
ΔG^0 (Hartree)	0.040	-0.010	0.031	0.003	0.039	0.004
ΔG^\ddagger (Hartree)	0.041	0.003	0.032	0.008	0.042	0.008
ΔG^\ddagger (Kcal/mol)	25.891	1.721	19.810	4.997	26.189	5.085

Reviewer 3-3: In the Scheme on page 54 of the SI, the representations of the radical Int-SI need to be connected by a mesomeric arrow, not by an equilibrium arrow. They are the same structure, just showing the spin densities on different centers. The paper is otherwise in very good shape, very detailed and will certainly attract the readers of Nature Communications. I strongly recommend acceptance after a final round of revisions.

Our response: Thank you for the valuable observation. We have revised the Scheme on page 54 of the Supporting Information accordingly.